# Comparing the effects of climate change labelling on reactions of the Taiwanese public

Li-San Hung 📷 [1✉] & Mucahid Mustafa Bayrak 📷 [1]

Scientists and the media are increasingly using the terms 'climate emergency' or 'climate crisis' to urge timely responses from the public and private sectors to combat the irreversible consequences of climate change. However, whether the latest trend in climate change labelling can result in stronger climate change risk perceptions in the public is unclear. Here we used survey data collected from 1,892 individuals across Taiwan in 2019 to compare the public's reaction to a series of questions regarding climate change beliefs, communication, and behavioural intentions under two labels: 'climate change' and 'climate crisis.' The respondents had very similar responses to the questions using the two labels. However, we observed labelling effects for specific subgroups, with some questions using the climate crisis label actually leading to backlash effects compared with the response when using the climate change label. Our results suggest that even though the two labels provoke similar reactions from the general public, on a subgroup level, some backlash effects may become apparent. For this reason, the label 'climate crisis' should be strategically chosen.

[1] Department of Geography, National Taiwan Normal University, 162, Section 1, Heping E. Rd., 106 Taipei City, Taiwan. ✉email: lshung@ntnu.edu.tw

Although some people and organisations have referred to anthropogenic climate change as an emergency or a crisis in the past[1–3], increasing scientific evidence[4] has convinced numerous scientists that our planet is facing a climate emergency[5]. Media outlets, such as the Guardian, have also announced that they prefer the terms 'climate emergency', 'crisis', or 'breakdown' over 'climate change' to reflect the current scientific reality more accurately[6]. The call from scientists and the media to change the terms used is aimed at urging the public, businesses, organisations, and governments to respond as quickly as possible to prevent the Earth's systems from reaching irreversible tipping points as a consequence of climate change[6,7]. The effects of using different terminologies, referred to as labelling effects, and the tailoring of information for specific purposes such as the promotion of public engagement, referred to as framing effects, are not new topics in climate change research. Many studies have shown that people react differently to the terms 'climate change' and 'global warming'[8,9], and several means of framing climate change, such as by referring to it as a public health issue[10] or a local issue[11], have been reported to significantly promote positive emotions or engagement with climate change. In terms of labelling effects, numerous studies have examined how the terms 'climate change' and 'global warming' influence people's perceptions and reactions to climate change. However, these studies have mainly been conducted in Western contexts[12–15]. Partisan differences in the usage and reactions to the abovementioned two terms in the United States have prompted many discussions[16–18]. In addition, racial and ethnic differences in labelling effects in the context of the United States have also been investigated[19]. Research on the labelling effects of increasingly-used terms, including 'climate emergency' and 'climate crisis', has been scant, with only one pilot study examining the labelling effects of the terms 'climate change', 'global warming', 'climate crisis', and 'climate disruption' among college students in the United States[20]. The results of that study suggested that 'climate crisis' not only evoked the worst reactions, but also showed backlash effects to these questions showed statistically significant differences. By contrast, 'climate disruption' evoked the most favourable reactions, with 'climate change' and 'global warming' responses located between the two extremes. Despite these informative results, our understanding of the labelling effects of these emerging terminologies on the broader population is limited. In addition, the labelling effects of climate change–related terms outside of the Western cultural context remain under-investigated.

Using the Taiwanese population as an example, this study compared the labelling effects between the terms 'climate change' and 'climate crisis' on multiple dimensions of climate change belief and behavioural intentions. We examined the labelling effects for 'climate crisis', not 'climate emergency', because 'climate crisis' is a more common term in the Taiwanese context as the Taiwanese public is more familiar with this term. The literal translation of 'climate crisis' in Chinese (chihouweiji) has been used in Taiwan for over 10 years. For example, this expression first appeared in a major local newspaper, the United Daily News, in 2007. By contrast, that newspaper only started to use the term 'climate emergency' (chihoujinji or chihoujinjijiuangtai) in 2019. Also, we chose 'climate change' instead of 'global warming' to be compared with 'climate crisis' because textbooks in Taiwanese school systems, including primary and secondary education, use 'climate change' as the thematic term to describe the changing global climate[21]. The current study systematically examined the labelling effects of the term 'climate crisis' on the public and provides empirical evidence to elucidate the labelling effects of climate change–related terms in a non-Western cultural context. Taiwanese society is heavily influenced by Confucian values[22] and

has traditionally been characterised by expressing high collectivistic values[23]. Furthermore, Taiwan is among the top countries in terms of per capita $CO_2$ emissions, both in East Asia and the world[24], and the Taiwanese government has set a goal of reducing the country's 2005 greenhouse gas emissions by 50% by 2050[25]. To achieve this goal, understanding the public's climate change risk perceptions and support of climate policy in Taiwan is imperative.

In addition to examining the labelling effects on the general population, we also divided our sample according to several key characteristics, which are central to the discussion of risk perception, as follows: gender (male and female), age (20–49 years and 50 years and above), educational attainment (high school diploma or below and some college or above), and cultural worldview (hierarchical, egalitarian, individualistic, and fatalistic values). Differences among these groups could improve understanding of how sociocultural factors affect labelling effects[26,27].

Our data were derived from a large telephone survey ($N$ = 1892) conducted across Taiwan in November 2019. Among the questions asked, participants answered 13 questions, revised or adopted from previous studies[28–31], regarding climate change beliefs, issue involvement, behavioural intentions, moral obligations, communication, and preferred societal responses with the random use of either the 'climate change' or 'climate crisis' label. The between-subject research design ensured that each participant received the same questions, except for the assigned label. In addition, participants' choice among the four statements regarding their views on nature indicated their cultural worldview[32]. Finally, basic demographic information was collected. The items used in the analysis are described in the methods section. Among the 1892 participants, 934 (49.3%) received questions using 'climate change' (hereafter 'climate change label' sample) and the remaining participants (958, 50.6%) were asked questions using 'climate crisis' (hereafter 'climate crisis label' sample). No differences were observed between the climate change label sample and the climate crisis label sample in terms of gender distribution, age structure or regional distribution.

## Results

We conducted a series of two-sided independent-samples $t$-tests to identify differences in labelling effects between the climate change label sample and the climate crisis label sample for both the full sample and subgroups (gender, age, educational attainment and cultural worldview). The 13 dependent variables were not normally distributed; however, it has been suggested that the results of $t$-tests are still robust under non-normally-distributed data, especially when the samples are large[33,34]. As a result, our discussion below was based on the results of $t$-tests, but we reported two cases that show significantly different results between $t$-tests and the Mann–Whitney tests in the methods section. For the full sample, the results revealed no statistically significant differences between the two label groups for all 13 questions (all $p \geq 0.067$, Table 1), suggesting that the terms 'climate change' and 'climate crisis' elicited similar responses in people for several aspects of climate change beliefs and engagement. We then considered the labelling effects for sociocultural factors. No differences were found in labelling effects between the two labels in terms of educational attainment, age or the holding of egalitarian values. By contrast, significant differences were observed for several questions in terms of gender and the holding of hierarchical, individualistic or fatalistic worldviews (Table 2; the full results are reported in supplementary information). Male respondents reported significantly lower mean frequency on three communication-related questions when the 'climate crisis' label was used than when the 'climate change' label was used

**Table 1 Two-sided independent sample *t*-test results of labelling effects for the whole sample.**

| Construct | Label | N | Mean(s.d.) | P values | Effect size (r) |
|---|---|---|---|---|---|
| Communication – discuss with family members | Change | 923 | 2.4 (1.08) | 0.091 | 0.04 |
| | Crisis | 947 | 2.32(1.04) | | |
| Communication – discuss with friends | Change | 932 | 2.5 (1) | 0.067 | 0.04 |
| | Crisis | 954 | 2.41(1) | | |
| Communications – information received | Change | 914 | 2.91(1.04) | 0.071 | 0.04 |
| | Crisis | 937 | 2.83(1.07) | | |
| Belief – personal harm | Change | 874 | 3.15(1.16) | 0.247 | 0.03 |
| | Crisis | 898 | 3.21(1.12) | | |
| Belief – harm future generations | Change | 885 | 4.03(0.93) | 0.549 | 0.01 |
| | Crisis | 901 | 4.05(0.92) | | |
| Belief – worry | Change | 919 | 4.05 (1) | 0.57 | 0.01 |
| | Crisis | 940 | 4.07(0.94) | | |
| Belief – personal importance | Change | 917 | 4.36(0.84) | 0.87 | 0 |
| | Crisis | 945 | 4.37(0.82) | | |
| Behavioural intention | Change | 904 | 4.3(0.83) | 0.578 | 0.01 |
| | Crisis | 935 | 4.32(0.8) | | |
| Involvement – priority for governments | Change | 884 | 2.93(0.82) | 0.179 | 0.03 |
| | Crisis | 900 | 2.88(0.85) | | |
| Belief – personal moral obligations | Change | 914 | 4.24(0.98) | 0.194 | 0.03 |
| | Crisis | 930 | 4.3(0.87) | | |
| Belief – moral obligations for future generations | Change | 919 | 4.4(0.84) | 0.842 | 0 |
| | Crisis | 936 | 4.41(0.8) | | |
| Belief – collective efficacy | Change | 889 | 4.28(0.96) | 0.856 | 0 |
| | Crisis | 923 | 4.27(0.97) | | |
| Involvements – climate issue important for voting decision | Change | 870 | 3.34(1.41) | 0.842 | 0 |
| | Crisis | 891 | 3.35(1.39) | | |

**Table 2 Two-sided independent sample *t*-test results of labelling effects for the subgroups.**

| Construct | Label | N | Mean(s.d.) | P values | Effect size (r) |
|---|---|---|---|---|---|
| **Gender-male** | | | | | |
| Communication – discuss with family members | Change | 456 | 2.31(1.08) | 0.023 | 0.08 |
| | Crisis | 440 | 2.15(1.01) | | |
| Communication – discuss with friends | Change | 462 | 2.5(0.98) | 0.008 | 0.09 |
| | Crisis | 444 | 2.32(1.02) | | |
| Communications – information received | Change | 449 | 2.88(1.01) | 0.002 | 0.1 |
| | Crisis | 432 | 2.67(1.04) | | |
| **Gender-female** | | | | | |
| Behavioural intention | Change | 455 | 4.31(0.78) | 0.018 | 0.08 |
| | Crisis | 499 | 4.42(0.66) | | |
| **Cultural worldviews – hierarchist** | | | | | |
| Belief – Collective efficacy | Change | 427 | 4.33(0.9) | 0.02 | 0.08 |
| | Crisis | 434 | 4.18(0.99) | | |
| **Cultural worldviews – individualist** | | | | | |
| Communication – discuss with family members | Change | 23 | 2.65(1.07) | 0.007 | 0.36 |
| | Crisis | 33 | 1.88(0.96) | | |
| Communication – discuss with friends | Change | 23 | 2.35(1.03) | 0.009 | 0.34 |
| | Crisis | 34 | 1.68(0.84) | | |
| **Cultural worldviews - fatalist** | | | | | |
| Belief – worry | Change | 45 | 3.38(1.21) | 0.03 | 0.24 |
| | Crisis | 36 | 3.94(1.09) | | |

(discussion with family members, crisis label $M = 2.15$ vs. change label $M = 2.31$, $p < 0.05$; discussion with friends, crisis label $M = 2.32$ vs. change label $M = 2.5$, $p < 0.01$; information received, crisis label $M = 2.67$ vs. change label $M = 2.88$, $p < 0.01$). Female respondents reported significantly higher mean in behavioural intention to engage in mitigation behaviour when the 'climate crisis' label was used than when the 'climate change' label was used (crisis label $M = 4.42$ vs. change label $M = 4.31$, $p < 0.05$). For cultural worldview, significant differences were observed for the following cases: (1) people with hierarchical values had a significantly lower mean in sense of collective efficacy when the

'climate crisis' label was used than when the 'climate change' label was used (crisis label $M = 4.18$ vs. change label $M = 4.33$, $p < 0.05$) and (2) people with individualistic values reported a significantly lower mean frequency of discussing climate change–related issues when the 'climate crisis' label was used than when the 'climate change' label was used (discussion with family members, crisis label $M = 1.88$ vs. change label $M = 2.65$, $p < 0.001$; discussion with friends, crisis label $M = 1.68$ vs. change label $M = 2.35$, $p < 0.01$). Because male respondents also reported a significantly lower frequency in climate change communication when the 'climate crisis' label was used than when the 'climate

change' label was used, we suspected that more men have individualistic values than do women. This suspicion was confirmed by an $X^2$ test, which revealed a significant association between gender and cultural worldview ($X^2$ (3) = 11.172, $p = 0.011$), and significantly more men (64%) than women (36%) expressed individualist values. Finally, people with fatalistic values were significantly more worried, shown in the mean, regarding the influence of climate change when the 'climate crisis' label was used than that when the 'climate change' label was used (crisis label $M = 3.94$ vs. change label $M = 3.38$, $p < 0.05$).

## Discussion

Similar to the results of a study on the labelling effects of the terms 'climate change' and 'global warming' in Americans and Europeans[13], our study revealed no major differences in labelling effects between the 'climate change' and 'climate crisis' labels. However, labelling effects were observed in male and female participants and in those with hierarchical, individualistic, and fatalistic worldviews, suggesting that the term 'climate crisis' should be used strategically[16]. Our finding that men less frequently engage in climate communication and obtain climate change information for the 'climate crisis' label than for the 'climate change' label are consistent with risk communication research that has indicated that men are less likely to act under negative frames[35]. Identifying the reasons for this behaviour among men requires further investigation, but identity-protective cognition[36], in which men express risk scepticism because of their individualistic worldviews, provides a possible explanation. In addition, the finding that women exhibited greater behavioural intention to mitigate climate change when the 'climate crisis' label was used than when the 'climate change' label was used is similar to the gender differences observed for climate change[27,37], which revealed positive effects for using the term 'climate crisis'. In terms of labelling effect differences according to worldview, a backlash effect was observed in people with hierarchical worldviews when the 'climate crisis' label[20] was used because they had a lower sense of collective efficacy for the 'climate crisis' label. The backlash effects are possibly attributable to a tendency of people with hierarchical worldviews to not trust others[38], which would result in a lower sense of collective efficacy. These backlash effects are particularly relevant in the high-collectivism Taiwanese society[39]. Collective framing, such as collective efficacy or collective responsibility, is a critical factor for Taiwanese people's motivation and engagement[24,40].

Similar to the aforementioned effects in men, people with individualistic values reported less frequent climate communication when the 'climate crisis' label was used, which could also be explained by identity-protective cognition[36] because they questioned the scientific labelling of 'crisis'. Finally, contrary to the findings of other studies that people with fatalistic values are typically unconcerned with matters outside of their control and show lower risk perceptions[41,42], our results indicated that they were actually more worried about the influence of climate change when the 'climate crisis' label was used. This may be because fatalistic views manifest differently in Asian and Western cultures[43], but it could also be because the 'climate crisis' label heightens some aspects of climate change perception (compared with the 'climate change' label), which enhances climate change engagement in people with fatalistic values. This finding illuminates an area that requires further research.

Overall, our results on the labelling effects between the terms 'climate change' and 'climate crisis' might disappoint people such as scientists and climate change activists because the 'climate crisis' label might not help to change the public's attitudes towards and engagement with climate change. Although we did find some positive labelling effects for specific subgroups, such as women, we also observed negative, backlash effects for other subgroups, such as people with hierarchical worldviews. Because no single terminology has identical effects on all people, identifying meaningful subgroups of the population[19,44] and understanding the labelling effects for these groups are paramount for audience-specific climate change risk communication.

Some critics of the 'climate crisis' and 'climate emergency' labels[1] have argued that these terms might imply that climate change solutions are led by governments instead of the public, that climate change concerns are prioritised over other social, cultural, and environmental issues, and that they might be counterproductive because they could reduce people's sense of efficacy. We did not find evidence of these disadvantages in our full sample because the public's reactions to the two labels did not differ in terms of governmental priorities, behavioural intentions, moral obligations, collective efficacy, or voting behaviours. We did observe the backlash effect in our subgroup samples, which further highlights the importance of targeted and tailored labelling and framing in climate change communication[14,45].

This study contributes to understanding of the labelling effects of the term 'climate crisis' in an under-investigated non-Western cultural context. Although place-specific differences exist, our study in Taiwan provide a critical understanding of labelling effects of climate-related terms in East Asian cultural contexts or other collectivist societies. In addition to exploring the affective dimensions of labelling effects, as was done in this study, more studies should investigate the cognitive dimensions and how the increasingly used terms influence people's certainty of the existence of climate change[18]. Also, the relations among people's experiences of extreme weather events, the availability heuristic, and the labelling effects are worth investigating[46–48]. Furthermore, future research could examine labelling effects among people in different cultural contexts. In countries such as the United States, examining the labelling effects of terms such as 'climate crisis' or 'climate emergency' in people with different political affiliations should yield valuable insights. Finally, the comparison of the increasingly used climate-related terms to 'global warming' deserves more attention in future studies.

## Methods

**Ethics statement**. This study was approved by the Research Ethics Committee of National Taiwan Normal University (No.: 201810HS018). Participants provided verbal consent before trained interviewers initiated each survey.

**Sampling method**. Our sample was derived from a large landline-based survey, conducted between 7 and 14 November, 2019, that targeted Taiwanese citizens who were at least 20 years old. The survey was performed by the Global View Survey Research Centre, a Taiwanese research company. We used stratified sampling based on the regions of Taiwan (22 cities and counties). Participants were randomly selected to answer questions that used either the 'climate change' or 'climate crisis' label. Overall, 1892 surveys were completed. Among the full sample, 934 (49.4%) participants answered questions that used the 'climate change' label and 958 participants (50.6%) answered questions that used the 'climate crisis' label. The response rate was 37.82%. The two sample groups were not different in terms of gender ($X^2$(1) = 1.6, $p = 0.205$), age ($X^2$(12) = 8.94, $p = 0.71$), educational attainment ($X^2$(6) = 1.74, $p = 0.94$), or regional distribution ($X^2$(21) = 2.41, $p = 1$). The two groups also did not statistically differ from the Taiwanese population in terms of gender structure (climate change label: $X^2$(1) = 0.06, $p = 0.81$; climate crisis label: $X^2$(1) = 2.41, $p = 0.12$) or regional distribution (climate change label: $X^2$(21) = 1.5, $p = 1$; climate crisis label: $X^2$(21) = 3.25, $p = 1$) but did significantly differ from the Taiwanese population in terms of age structure (climate change label: $X^2$(12) = 62116.08, $p < 0.001$; climate crisis label: $X^2$(12) = 80725.11, $p < 0.001$). In comparison with the Taiwanese population, the sample contained fewer young people and more older adults, a phenomenon often seen in telephone surveys[49]. We used unweighted samples for analysis.

**Questionnaire development**. This study used 13 questions in the survey that was specifically designed to examine the labelling effects for the terms 'climate change' and 'climate crisis'. All participants were asked the same questions in the same

order, except for random use of the terms 'climate change' and 'climate crisis'. We used a between-subject design, which means each participant answered questions phrased with the same label (i.e., 'climate change' or 'climate crisis') throughout his or her survey. The survey questions were adopted or revised versions of questions from related studies[28–31]. Shortened versions of the 13 questions are cited in the main text. The survey included the following questions:

(1) Communication (discussion with family members). How often do you talk to your family members about issues related to climate change (the climate crisis)?
(2) Communication (discussion with friends). How often do you talk to your friends about issues related to climate change (the climate crisis)?
(3) Communication (information received). How often do you inform yourself regarding the harm to Taiwan from climate change (the climate crisis)?
(4) Belief (personal harm). How much do you think climate change (the climate crisis) will harm you personally?
(5) Belief (harm future generations). How much do you think climate change (the climate crisis) will harm future generations?
(6) Belief (worry): How worried are you about the influence of climate change (the climate crisis)?
(7) Belief (personal importance). How important is the issue of climate change (the climate crisis) to you personally?
(8) Behavioural intention. I plan to take steps to reduce my influence on climate change (the climate crisis). Do you agree?
(9) Involvement (priority for governments). Do you think that climate change (the climate crisis) should be a low, medium, high, or very high priority for the central government and Executive Yuan?
(10) Belief (personal moral obligations). Some people say 'I think I have the moral obligation to reduce my impact on climate change (the climate crisis)'. Do you agree?
(11) Belief (moral obligation to future generations). Some people say 'I think I have the moral obligation to reduce my impact on climate change (the climate crisis) for future generations'. Do you agree?
(12) Belief (collective efficacy). Some people say 'If we work together, we can reduce the threat of climate change (the climate crisis) to humans.' Do you agree?
(13) Involvement (climate issue relevance to voting decision). Taiwan will elect its president on 11 January, 2020. Do you agree that the climate change (the climate crisis) issue is an important consideration for your decision regarding which candidate to vote for? All questions, except for question nine, were measured using a five-point Likert scale, where "1" signifies the least favourable response and "5" signifies the most favourable response. For question nine, a 4-point scale was used, following the same scoring logic. For questions regarding cultural worldview, we adopted a question used in a previous study[32] that asks respondents to choose the statement that most accurately represents their views regarding environmental problems from the following four statements: (1) environmental problems can only be controlled by enforcing radical changes in human behaviour and in society as a whole (egalitarian values); (2) environmental problems are not out of control, but the government should dictate clear rules regarding what is and what is not allowed (hierarchical values); (3) we do not need to worry about environmental problems because these problems will ultimately always be resolved through technological solutions (individualistic values); and (4) we do not know whether environmental problems will intensify (fatalistic values). The four choices for cultural worldview were presented in a random order. The translation from Chinese and English was performed by the first author and was reviewed by survey experts from the Global View Survey Research Center. The sample sizes, means, and standard deviations for each of the variables for the 'climate change' and 'climate crisis' labels are listed in Table 1.

**Data management and analysis**. The survey data were managed using SPSS version 23[50]. We used two-sided independent-samples t-tests to investigate the labelling effects between the two groups. We calculated the effect sizes for t-tests by the following equation[51]: $r = \sqrt{\frac{t^2}{t^2 + df}}$. The results of the t-tests that examined the labelling effects on the full data set are provided in Table 1. The full results of the t-tests that examined the labelling effects by gender, age, educational attainment and cultural worldview are provided in Tables 1, 2, 3, and 4, respectively, in the supplementary information. As discussed in the main text, we were aware that the 13 outcome variables were not normally distributed. Because it has been suggested that the application of t-tests does not require the assumption of normal distribution, especially for large samples[33,34], we kept the parametric method in our analysis. Nevertheless, we conducted Mann–Whitney tests for all the analysis, and found two cases where the statistical significance was different, at $p = 0.05$ level, between the two tests. The first case was behavioural intention for female subgroups, where in t-test it was statistically significant ($p = 0.018$, $r = 0.08$), but in Mann–Whitney test it was not significant (asymptotic significance $p = 0.061$, $r = 0.06$). The second case was collective efficacy for people with equalitarian

worldviews, where in t-test it was not significant ($p = 0.059$, $r = 0.07$), but in Mann–Whitney test it was significant (asymptotic significance $p = 0.049$, $r = 0.07$). The statistically significant result suggested that for people with equalitarian worldviews, the climate crisis label result in stronger sense of collective efficacy than the climate change label (crisis label $M = 4.44$, Mdn = 5 vs. change label $M = 4.32$, Mdn = 5, $U = 82,900.5$, $z = 1.971$, $p = 0.049$, $r = 0.07$). The effect sizes for Mann–Whitney tests were calculated by the following equation[51]: $r = \frac{z}{\sqrt{N}}$. The effect sizes for two-sided independent-samples t-tests and Mann–Whitney tests were calculated in Microsoft Excel version 2016.

**Reporting summary**. Further information on research design is available in the Nature Research Reporting Summary linked to this article.

## Data availability
The data that support the findings of this study are available from the corresponding author upon reasonable request.

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

## Acknowledgements
This study is supported by the Ministry of Science and Technology (MOST) of Taiwan under Grant No. MOST 108-2636-H-003-004 and MOST 109-2636-H-003-004.

## Author contributions
L.S.H. designed the survey used within the work presented, with M.M.B. providing suggestions to the survey. L.S.H and M.M.B. organised and managed the data. L.S.H. analysed the data. L.S.H. and M.M.B. interpreted the results. L.S.H. led in writing the paper, developing this with input from M.M.B.

## Competing interests
The authors declare no competing interests.
