## [Peer Review File · Nature Communications]

REVIEWER COMMENTS

Reviewer #1 (Remarks to the Author):

Thank you for this opportunity to review the manuscript submitted to Nature Communications titled, "Climate change or crisis? Comparing the effects of climate change labelling on reactions of the Taiwanese public." The paper reports on a survey experiment among the Taiwanese public to test the effect of "climate change" vs. "climate crisis" labeling--an important practical matter, because as the authors point out, journalistic norms have recently shifted in the direction of "crisis" labeling, and limited research exists on this question (as compared to the more studied "global warming" vs. "climate change" comparison). Overall, the research finds little evidence that this labeling difference matters across a wide range of attitudinal and behavioral survey measures. In general, the work appears to be useful to scholars and practitioners interested in climate change communication (and labeling effects specifically).

Although I think this is a useful study that should be published in an academic journal, it seems to be a better fit for a specialty journal than Nature Communications, where these matters are more often discussed and where this work might better reach the audience that regularly engages with these questions (such as Climatic Change and Journal of Environmental Psychology, for example). As feedback, I might encourage the authors to consider collecting comparative data that might speak to how different cultural/political contexts might moderate this labeling effect--this might be especially interesting and important, given that a UK newspaper (The Guardian) famously implemented the "climate crisis" label. To be sure, I'm not arguing that this study *needs* Western survey data to be publishable, only that some further development of cultural/political context and its possible role might add a lot of value to what is already a strong piece of research.

Reviewer #2 (Remarks to the Author):

Revision: Climate change or crisis? Comparing the effects of climate change labelling on reactions of the Taiwanese public

General Comments

An interesting analysis of the paper is to measure the perception on "climate change" and "climate crisis". Given the findings, "climate crisis" should be strategically chosen. This is not a new topic of research. However, the authors made some important efforts on this current subject.

There is a superficial justification on the choose for "climate change" and "climate crisis".

One of main justification of this study is to understand the Western cultural context. To study Taiwan, particularly is important given the higher levels of CO2 emissions per capita. However, there is no a greater novel in the current version of this study.

Despite data support the conclusions, there are no theoretical advances and not expressive empirical novels.

Minor comments

Line 43 – climate emergence and climate change as novel terms. I believe that these terms cannot be described as new in the context in question. Please see, Archer and Rahmstorf (2010) and Crist (2007).

Archer, D., & Rahmstorf, S. (2010). The climate crisis: An introductory guide to climate change. Cambridge University Press.

Crist, E. (2007). Beyond the climate crisis: A critique of climate change discourse. *Telos*, 141 (Winter), 29-55.

Line 73 – Why the age was divided in this way? It is not clear to me. There is a difference of almost 30 years between 20-49. Probably a young person (e.g. 20 years old) has a very different perception on climate from a 49 person who may even be your father/mother. This division can influence the results of t tests.

Line 105 - Male respondents reported significantly lower frequency on three communication-related questions when the 'climate crisis' label was used than when the 'climate change' label was used. What means the term frequency here? Reading in the sequence, it seems that it is the mean.

Line 111- Along the text the authors describe the term "intention". However, there is no clear how intention was measured. A large body of researchers analyzed intention through theory of planned behavior, proposed by Icek Ajzen. It is clear to me that this approach was not used here. But what approach was used?

Line 189 – Future studies also should analyze people perception on climate change in regions with recent extreme climate events and other without these events. Thus should be measured if the availability heuristic (see the book *Think, fast and slow* – Daniel Kahneman).

Line 292 – As t test was used it is assumed that the data distribution is normal. I suggest that this information be presented in the text.

Lines 97 – after see Table 1 I think that the correct is $p \geq .067$.

Reponses to reviewers' comments [NCOMMS-20-16061-T]

Reviewer #1 (Remarks to the Author):

Thank you for this opportunity to review the manuscript submitted to Nature Communications titled, "Climate change or crisis? Comparing the effects of climate change labelling on reactions of the Taiwanese public." The paper reports on a survey experiment among the Taiwanese public to test the effect of "climate change" vs. "climate crisis" labeling--an important practical matter, because as the authors point out, journalistic norms have recently shifted in the direction of "crisis" labeling, and limited research exists on this question (as compared to the more studied "global warming" vs. "climate change" comparison). Overall, the research finds little evidence that this labeling difference matters across a wide range of attitudinal and behavioral survey measures. In general, the work appears to be useful to scholars and practitioners interested in climate change communication (and labeling effects specifically).

Response:

First, we would like to thank Reviewer #1 very much for providing the valuable comments. We also appreciate that you found this study important to scholars and practitioners interested in climate change communication. We truly appreciate the opportunity to revise our paper, and believe that our manuscript has significantly improved based on the comments provided by the reviewers.

*Although I think this is a useful study that should be published in an academic journal, it seems to be a better fit for a specialty journal than Nature Communications, where these matters are more often discussed and where this work might better reach the audience that regularly engages with these questions (such as Climatic Change and Journal of Environmental Psychology, for example). As feedback, I might encourage the authors to consider collecting comparative data that might speak to how different cultural/political contexts might moderate this labeling effect--this might be especially interesting and important, given that a UK newspaper (The Guardian) famously implemented the "climate crisis" label. To be sure, I'm not arguing that this study *needs* Western survey data to be publishable, only that some further development of cultural/political context and its possible role might add a lot of value to what is already a strong piece of research.*

Response:

As more and more scholars, businesses, and news media start to use the term “climate crisis” we believe that the content of this manuscript – comparing the labeling effects between “climate crisis” and “climate change” – merits publication in *Nature Communications*, which is not only a transdisciplinary journal but it also targets a broad academic and non-academic (e.g. the media, practitioners, or policymakers) audience. Hence, we strongly believe that we can influence both the academic and public debates on this topic by publishing this article in *Nature Communications*. In addition, we have added a discussion on how Taiwanese society has been culturally influenced by Confucianism and collectivism in the first half of our manuscript (lines 70-72). This helps the reader better understand our discussion on the framing effects of collective efficacy for Taiwanese society (lines 160-168).

Reviewer #2 (Remarks to the Author):

Revision: Climate change or crisis? Comparing the effects of climate change labelling on reactions of the Taiwanese public

General Comments

1. An interesting analysis of the paper is to measure the perception on “climate change” and “climate crisis”. Given the findings, “climate crisis” should be strategically chosen. This is not a new topic of research. However, the authors made some important efforts on this current subject.

Response:

First, we would like to thank Reviewer#2 very much for providing the valuable comments. We truly appreciate the opportunity to revise our paper, and believe that our manuscript has significantly improved based on the comments provided by the reviewers.

2. There is a superficial justification on the choose for “climate change” and “climate crisis”.

Response: The reason why we compared the term “climate change” with “climate crisis” is because “climate change” is the thematic and most common term for describing the global changing climate in the textbooks of Taiwanese primary and secondary school systems. Hence, Taiwanese people are more familiar with the Chinese term for climate change instead of global warming. We elaborated on this in

lines 64-67. We further include a sentence on the comparison of increasingly used climate-related terms (e.g., climate crisis, climate emergency) with “global warming” as a future research topic (lines 213-215). In addition, we chose “climate crisis” over “climate emergency” because it is also a term that the Taiwanese public are more familiar with (lines 57-64).

3. One of main justification of this study is to understand the Western cultural context. To study Taiwan, particularly is important given the higher levels of CO2 emissions per capita. However, there is no a greater novel in the current version of this study.

Response: We believe that our Taiwanese case provides a critical understanding of labeling effects of climate change related terms in East Asian cultural contexts other collectivist societies, and other cultural contexts (including the Western context). We added this discussion in lines 202-205. In addition, our study provides important insights into the effectiveness of using different climate change labelling terms, which are important for cross-country studies and analysis.

4. Despite data support the conclusions, there are no theoretical advances and not expressive empirical novels.

Response: As the labeling effects of increasingly-used climate-change-related terms (e.g., climate crisis and climate emergency) have not yet been systematically examined, we believe that this manuscript does advance people’s understanding on labelling effects, especially in the East-Asian context. In addition, this study also has important policy-relevant implications. In sum, we believe that this study does contribute to current academic understandings of labeling effects in addition to its empirical contributions.

Minor comments

Line 43 – climate emergence and climate change as novel terms. I believe that these terms cannot be described as new in the context in question. Please see, Archer and Rahmstorf (2010) and Crist (2007).

Archer, D., & Rahmstorf, S. (2010). The climate crisis: An introductory guide to climate change. Cambridge University Press.

Crist, E. (2007). Beyond the climate crisis: A critique of climate change discourse. Telos, 141 (Winter), 29-55.

Response: We agree with this comment that these terms should not be described as “new” terms. We changed it to “increasingly used” terms or “emerging” terms. All related parts have been modified throughout the manuscript. The two references are also cited in the text. Thank you for suggesting these studies to us.

Line 73 – Why the age was divided in this way? It is not clear to me. There is a difference of almost 30 years between 20-49. Probably a young person (e.g. 20 years old) has a very different perception on climate from a 49 person who may even be your father/mother. This division can influence the results of t tests.

Response: We divided age variable into 20- 49 years-old as well as 50 years and above because this was the best way that divided the sample into two groups with similar sample sizes. The age variable was measured by several 5-year range choices: 20-24 years-old, 25-29 years-old, ..., and 75 years-old and above. By dividing age variable in this way, the 20-49 years-old accounted for 45.8% of sample ($n = 863$), while the 50 years-old accounted for 54.2% of the sample ($n = 1022$).

We tried to divide age into 20-39 years-old ($n=513$, 27.2%) and 40 years-old ($n = 1372$, 72.8%), and the results of independent t-tests suggested that there were no statistically significant differences between the two label terms for all 13 questions in both age groups (all $p \geq 0.54$).

Line 105 - Male respondents reported significantly lower frequency on three communication-related questions when the ‘climate crisis’ label was used than when the ‘climate change’ label was used. What means the term frequency here? Reading in the sequence, it seems that it is the mean.

Response: We have changed it from “lower frequency” to “lower mean frequency.” (line 118). We also revised other parts in the manuscript that had similar issues.

Line 111- Along the text the authors describe the term “intention”. However, there is no clear how intention was measured. A large body of researchers analyzed intention through theory of planned behavior, proposed by Icek Ajzen. It is clear to me that this approach was not used here. But what approach was used?

Response:

The intention variable was revised from Brody et al. (2012). The survey question was

“I plan to take steps to reduce my influence on climate change (climate crisis). Do you agree?” We now refer to the sources of the survey questions in the main text (line 86). Detailed information regarding methods are discussed in the methods section.

Brody S, Grover H, Vedlitz A (2012) Examining the willingness of Americans to alter behaviour to mitigate climate change. *Climate Policy* 12:1–22.

<https://doi.org/10.1080/14693062.2011.579261>

Line 189 – Future studies also should analyze people perception on climate change in regions with recent extreme climate events and other without these events. Thus should be measured if the availability heuristic (see the book Think, fast and slow – Daniel Kahneman).

Response: Thank you for the suggestion. We have added this as a future research topic. (lines 208-210).

Line 292 – As t test was used it is assumed that the data distribution is normal. I suggest that this information be presented in the text.

Response:

We are thankful for this valuable comment. We were aware that the 13 outcome variables were not normally distributed. It has been suggested that t-tests are still robust under the non-normally-distributed data, especially when the sample sizes are large (Poncet et al. 2016; Lumley et al., 2002). As a result, we keep the parametric method for this study. We did conduct the Mann-Whitney tests for all the analysis, and found that most of the results are the same in terms of statistical significance (at $p = .5$ level), and only 2 cases showed a new statistical significance. We reported these two cases in detail in the methods section. More importantly, our main conclusions – including (1) no framing effects for the full sample; (2) framing effects occur in the subgroup samples, and (3) backlash effects are shown for specific subgroup cases – did not change when employing both statistical methods.

The above discussion appears in the main text (lines 103-108) and methods section (lines 309-324)

Lines 97 – after see Table 1 I think that the correct is $p \geq .067$.

Response:

Thank you for spotting this error. We have corrected this error (line 109).

REVIEWERS' COMMENTS:

Reviewer #2 (Remarks to the Author):

Reading the comments of authors I believe that the manuscript was improved. My csuggestion were complished in full. The paper present relevant contributions for academic debate. I would like to highlight that, while the data analysis are simply (mean test),simplicity is important for advances in science.

Response to reviewers' comments [NCOMMS-20-16061A]

Reviewer #2 (Remarks to the Author):

Reading the comments of authors I believe that the manuscript was improved.

My suggestion were accomplished in full. The paper present relevant contributions for academic debate. I would like to highlight that, while the data analysis are simply (mean test), simplicity is important for advances in science.

Response. We are thankful for reviewer#2's comments, and we also believe that our manuscript is greatly improved because of reviewers' comments.